# Navigating cancer: Insights from patient journey mapping

Sarah Day[1]*, Jane Harries[2], Bukeka Sawula[1], Alec Payne[1], Shameem Bray[1], Denis Okova[1], Lauren Pretorius[3], Jennifer Moodley[1]

1 Division of Public Health Medicine, School of Public Health, Faculty of Health Sciences, University of Cape Town, Cape Town, South Africa, 2 Cancer Association of South Africa, Cape Town, South Africa, 3 Campaigning for Cancer, Johannesburg, South Africa

* sarah.day@uct.ac.za

## Abstract

### Background

Cancer is an increasing public health problem in South Africa, with breast cancer being the most diagnosed cancer and cervical cancer the leading cause of cancer deaths among women. Despite the complexity of breast and cervical cancer patients' journeys through the healthcare system, patients' voices are still predominantly missing from the body of literature. Patient journey mapping, as a qualitative research method, offers an opportunity for centring patients in their care journeys and reimagine healthcare provision for the potential improvement of health systems and patient outcomes.

### Aim

The aim of this study was to map journeys of breast and cervical cancer patients across the cancer care continuum.

### Methods

Using patient journey mapping, we conducted six focus group discussions with patients with breast and cervical cancer who had completed treatment in Gauteng, KwaZulu Natal and the Western Cape, South Africa. The process involved three steps: 1.) development of individual maps; 2.) narrative sharing; and 3.) development of a collective map. Results of the study were shared in feedback sessions.

### Findings

A total of 31 people participated in the focus groups: 23 with breast cancer, 7 cervical cancer and one had both cancers during her lifetime. The participants' ages ranged between 30 and 69 years old. A patient journey map was developed drawing on the individual and collective maps and participant narratives. The findings of

**Data availability statement:** The data from this study cannot be shared publicly due to ethical restrictions imposed by the University of Cape Town Faculty of Health Sciences Human Research Ethics Committee, as public deposition may compromise patient privacy. Data are available for researchers who meet the criteria for access to confidential data via the University of Cape Town Faculty of Health Sciences Human Research Ethics Committee (hrec-enquiries@uct.ac.za).

**Funding:** Cancer Association of South Africa (CANSA) Type B Grant. The funders had no role in study design, data analysis or decision to publish. The funder assisted with participant recruitment but was not present during data collection. Prof Jane Harries, who is the Head of Research at CANSA, provided input on the fully drafted manuscript.

**Competing interests:** I am an academic editor for PLOSOne. This does not alter our adherence to PLOS ONE policies on sharing data and materials. All other authors have no conflict of interest to declare. However, the data from this study cannot be shared publicly because public deposition would breach compliance with the protocol approved by your research ethics board, as it would compromise patient privacy. The data contains detailed accounts of patient journeys through the healthcare system, which may make them identifiable.

the paper constellate around three themes. The first theme, (de)personalised care, offers an examination of how relational, institutional and structural factors shape and are reshaped through participants lived experiences across the cancer care continuum. The second theme, self-advocacy, explores how participants advocate for their healthcare needs throughout the cancer care continuum. The third theme, intersecting vulnerabilities, explores how intersecting social identities, such as socioeconomic factors, gender, comorbidities and mental health, shape their cancer care journeys.

## Conclusions

By centring patient with breast and cervical cancer voices, patient journey mapping not only showed where services and systems fall short but also provided guidance for redesigning a more patient responsive health system.

## Introduction

Low-and-middle income countries (LMICs) face a disproportionate burden of cancer-related death [1]. Increasing cancer incidence and poorer prognosis in LMICs can be attributed to ageing societies, health disparities, poor access to quality healthcare, inequalities in timely care-seeking and high prevalence of risk factors [1]. Of the 13 million cancer deaths estimated in 2030, 75% will occur in LMICs. Most cancer patients in Southern Africa present with advance stage disease, which is associated with a poorer prognosis [2–4]. Cancer is an increasing public health problem in South Africa (SA), with breast cancer being the most commonly diagnosed cancer and cervical cancer the leading cause of cancer deaths among women [5]. In 2017, SA adopted Breast [6] and Cervical [7] Cancer Prevention and Control Policies. Both policies offer comprehensive guidelines for promoting community awareness, screening, primary and secondary prevention, early detection, diagnosis, treatment, post-treatment care, palliation and end of life care. These policies are supported by the National Cancer Strategic Framework 2017–2022 [8] and the National Policy Framework and Strategy on Palliative Care 2017–2022 [9].

In SA, as in most of the continent, patients with possible cancer symptoms usually self-present to primary healthcare facilities and are then referred to secondary and/or tertiary facilities for further diagnostic investigations and treatment. This journey to diagnosis and subsequent care is complex and influenced by multiple factors, including knowledge and awareness of cancer symptoms; perception of risk; the nature of the symptoms; availability of diagnostic investigations; trained staff; availability of treatment options and health system, psychological, socio-cultural and economic barriers to care [10–16]. Despite the complexity of breast and cervical cancer patients' journeys through the healthcare system, SA patients' voices are still predominantly missing from the body of literature.

Patient journey mapping, as a qualitative research method, offers an opportunity for centring patients in their care journeys and reimagining healthcare provision for the potential improvement of health systems and patient outcomes [17,18]. Providing

a visual representation of the route patients take at all phases of their care trajectory and their experiences across this journey [18], patient journey mapping can assist with identifying gaps, suboptimal processes, areas for improvement and current system strengths while overlaying embodied and spatial aspects of lived experiences of healthcare journeys [19]. Journey mapping is a ubiquitous term and refers to several different approaches and methodologies used to capture patients' healthcare journeys, including semi-structured interview, electronic health records, participatory action research approaches, observations, autoethnography and surveys [20]. Globally, patient journey mapping has been used to explore the care journey for patients with breast [18], cervical [21], colorectal [19], prostate [22,23], lung [24], ovarian [25] and skin [26] cancer among others. Such studies have taken place in Spain [18,26], United Kingdom [19,26], Finland, Germany, Greece, Italy, Malta, Poland, Romania, the Netherlands [26], Sweden [22], Japan [25] and Canada [27]. Although there are a few notable exceptions [17,27], LMICs and marginalised communities remain largely underrepresented.

This study is embedded within the cancer care continuum [28]. Understanding patients' cancer journeys across the continuum highlight disparities, and gaps in access and care [29]. While patient journey mapping has not been used in the South African context for patients' cancer journeys, other literature has documented barriers and facilitators experienced by patients [10,16,30,31]. The aim of this study was to map journeys of breast and cervical cancer patients across the cancer care continuum.

## Research design

This qualitative study drew on focus group discussions (FGDs) accompanied by patient journey mapping for breast and cervical cancer patients across the care continuum. The journey mapping process was adapted from Kelly et al., [32] and BC Cancer et al., [27] and took a whole person approach considering social and emotional wellbeing, family and community commitments, personal, spiritual, and cultural considerations, and physical and biological experiences. The mapping process was collaborative and visually captured complex journeys.

## Health system context

SA has a dual public and private health care system. The public sector services 71% of the population. The private sector – which services around 27% of the population – is funded through individual's contributions to medical aid schemes [33]. The South African public sector healthcare system is constituted by three tiers – primary, secondary and tertiary. Screening happens mostly at primary healthcare clinics. Patients who have possible breast and cervical cancer symptoms typically self-present at primary healthcare clinics and are seen by nurses or, less frequently, medical officers. Those with possible cancer symptoms or abnormalities are referred to secondary facilities (regional/district hospitals) for investigation. Depending on the health facility resources, patients will either be diagnosed and treated at the secondary facility, or they will be referred to a tertiary facility for diagnostic investigations and treatment. This study took place in three of nine provinces in South Africa viz. Gauteng, KwaZulu Natal and the Western Cape. These provinces represent differences in access to cancer care [34]. Data was collected between February and April 2024 with feedback sessions conducted between July and October 2024.

## Theoretical framework

The theoretical underpinnings of this research are intersectionality and health equity [35]. These conceptual coordinates provide a framework to understand power relations, social identities and systems of inequality within cancer care journeys [35]. An intersectionality health equity lens recognises the ways in which patients' social identities – such as race, gender, sexuality, class and disability – can adversely impact their cancer care journeys as they are embedded within intersecting systems of oppression and can shape their health outcomes [35–37]. Patients with cancer have often been dislocated from their social identities which creates a unidimensional view of their lived experiences [38–40]. Taking on an intersectional lens can shift towards a multidimensional, nuanced understanding of cancer patient's journey to diagnosis

and provide a lens through which to explore barriers to early cancer diagnosis and identify interventions that specifically address tacit barriers to care [36].

## Sample

We conducted patient journey mapping with people living with breast and cervical cancer who had completed treatment. Using convenience sampling, patients in both the public and private sectors were included. Participants were recruited through our partner organisations, the Cancer Association of South Africa (CANSA) and Campaigning for Cancer, through their existing patient advisory boards, care homes and support groups. We contacted relevant stakeholders at each organisation and requested the number of participants required, participant eligibility criteria and the proposed dates and venues for data collection. These stakeholders then invited potential participants to join the study.

## Patient journey mapping process

SD, BS, SB and AP conducted the FGDs in the Western Cape. SD and BS conducted the FGDs in Gauteng and KwaZulu-Natal, and a translator was employed to assist with translation. Data collection took place in a rented room at a public library in the Western Cape, and in the boardrooms of our NGO partners in Gauteng and KwaZulu-Natal.

**Pilot.** Following a pilot FGD in the Western Cape, we refined the patient journey mapping process combing participants with both cancers and separating participants undergoing palliative and end of life care. Originally, we planned for participants to co-create a map together on an A0 map of the province where the particular focus group discussion took place. During piloting, participants elected to use blank pieces of paper to map out their own journey individually, leaving the collective map untouched in the centre of the table. It was also challenging to provide geographic maps that would capture the participants' full journeys, with some participants moving between municipalities and provinces during their cancer care journey. We therefore adapted the process to start with an individual journey mapping process. We also removed the originally planned geographic map of the province in favour of listing headings (journey to diagnosis, getting a diagnosis, treatment and life after treatment) along the care continuum. The headings provided specific focus areas for a collective map making process at the end.

**Data collection.** FGD took up to four hours and were conducted by at least three trained facilitators, at least one of whom was familiar with the relevant local language. The FGDs involved three steps.

**Step 1: Development of individual maps.** Participants were provided with A3 size paper, coloured pens and were given 20 minutes to draw or write their journey from symptom identification or screening to the day they arrived at the FGD.

**Step 2: Narration of health journey.** After the completion of the individual maps, participants were asked to share a narrative account based off a single question: "Can you tell me about your health journey related to your cancer diagnosis?". Taking a semi-structure approach, the FGD guide was developed to prompt for intersecting identities and factors which shape patients' cancer care journeys if needed. Each participant was given space to tell their story (relatively) uninterrupted in a language of their choosing.

**Step 3: Creation of collective map.** To co-create the map, each participant used sticky notes to add their contribution under each heading on a larger paper. The sticky notes were read out and the participants were asked what needed to be added, which sparked further discussion which the facilitator transcribed onto the map. The collective mapping process stopped when all the participants were satisfied with what was included in the map.

## Feedback sessions

Results from the original FGDs were taken back to participants, either in-person or online, which allowed for participant reflection, correction and elaboration.

 

## Ethics

This study received ethical approval from the University of Cape Town Faculty of Health Sciences Human Research Ethics Committee (HREC Ref: 826/2023). All participants were asked required to provide written informed consent prior to the FGD. The study was explained in English and in local languages when needed. The FGDs took place at a location advised by partner organisations, such as local libraries and partner organisation offices. Participants selected their own pseudonyms. Participants were reimbursed R200 for their time and R150 for their travel. Lunch was provided for the participants. All transcripts were anonymised at the time of transcription, before being sent to the rest of the research team. Audio recordings and transcriptions were stored in a shared DropBox with the research team. Access to the DropBox folder is controlled by JM.

## Analysis

The data was analysed using interpretive phenomenological analysis (IPA). Drawing on IPA, as an analytic frame, this study offers a close examination of the lived experience of breast and cervical cancer patients' journeys across the care continuum [41]. This analysis considered individual patient journeys as embedded within context, as whole persons on a particular health journey, and identify barriers and enablers that impacted their journey [17]. Drawing on the theoretical coordinates of the study, the analysis focused on the ways in which participants' intersecting social identities shape their everyday embodied experiences of their cancer care journeys [35]. Specifically, we looked at the ways in which power and privileged shaped participants' lived experiences, and how participants (re)shaped these power dynamics. A phenomenological analysis of intersecting social identities allows for an examination of how patients are orientated within the cancer care continuum to institutions and other people, and how social identities become racialised, gendered and classed within social interactions at different points in the healthcare system [42].

IPA analysis involved (1) emersion in the data, (2) clustering the data into emergent themes, and (3) structuring the analysis. The research team (SD, AP, BS, SB, DO and JM) read through the transcripts, shared initial impressions and developed a list of codes. NVivo 14 was used for coding and two coders (SD, AP, BS, DO) inductively developed codes in each transcript. Throughout the coding process, possible new codes were discussed and added to a shared online code book. When new codes were added, the coders returned to previously coded transcript to ensure the new code has been added. After coding, emergent themes were developed, noting similarities and differences between the FGDs. The emergent themes were then grouped into superordinate themes through team discussions, during which SD collaborated with BS, AP, and JM to refine and finalize the thematic structure. The emergent themes identified through the IPA process were translated into easy-to-understand visual representations of cancer care journeys across the continuum for breast and cervical cancer patients as well as a written document.

## Findings

A total of six FGDs were conducted – three in the Western Cape, two in Gauteng and one in KwaZulu Natal – with a total of 31 participants: 23 with breast cancer, 7 cervical cancer and one had both cancers during her lifetime. The FGDs were conducted between February and April 2024 with feedback sessions conducted between July and October 2024. The participants' ages ranged between 30 and 69 years old. Eighteen participants exclusively utilised public healthcare services, 7 relied solely on private healthcare and 6 accessed both public and private healthcare services. Some participants moved between sectors. Dates that participants were first diagnosed ranged from 2000 to 2023. Three participants have passed away since participating in this study.

### Overview of journey

Fig 1 below depicts patient journeys through the cancer care continuum. Most participants self-identified symptoms as requiring medical attention, regardless of whether they attributed them to symptoms of cancer. While many participants

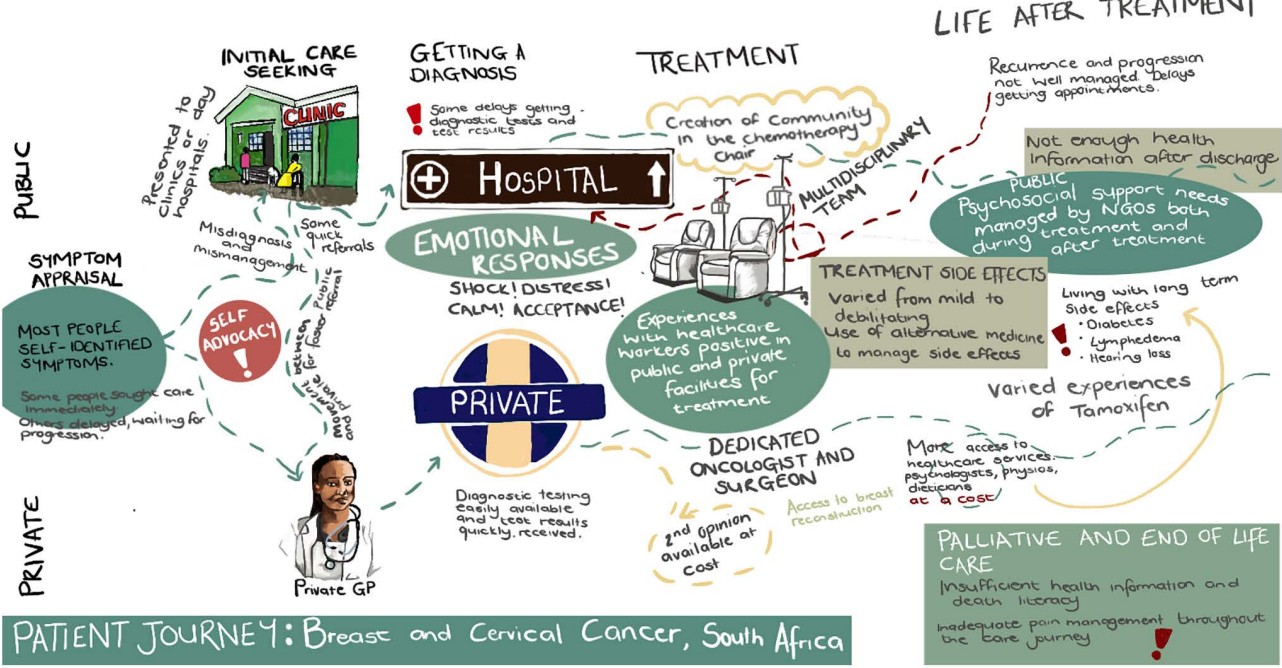

**Fig 1. Composite patient journey map.**

did not delay care seeking, some misappraised their symptoms to be related to benign conditions, infections, menopause, pregnancy, comorbid health conditions. Some delayed for fear of a cancer diagnosis. Participants had different points of entry into the healthcare system – with public sector participants using clinics or day hospitals as an initial point of entry and private sector participants using private general practitioners (GPs) to bypass straight to specialists. Some participants used private GPs as entry points into the healthcare system or when they experience suboptimal PHC care before being referred to the public sector for diagnostics and/or treatment. Delays to diagnosis were predominantly reported from participants in the public healthcare sector, due to misdiagnosis, loss of folders, refusal to refer, and inappropriate management of symptoms. Conversely, participants who were in the private sector experienced much faster journeys to diagnosis. Pathways to diagnosis were faster for patients with breast cancer as compared with cervical cancer. Participants had mixed reports about how their diagnosis was conveyed to them by HCWs. Some participants reported humanising experiences when getting their diagnosis, while others stated that they were moved between facilities without clear communication around their diagnosis. Participants in both the public and private sector predominantly had positive experiences with the health facilities where they received treatment. Participants in the public sector reported different waiting times to start treatment – while some started treatment quickly, others had to wait extended periods prior to the start of treatment. Conversely, in the private sector, participants reported quick journeys from diagnosis to treatment initiation.

Participants had mixed experiences surrounding follow-up care and access to relevant post-treatment health information in the public sector, with some reporting ease in ability to access follow-up care and others reporting that they felt "abandoned" after discharge and relied on NGOs for psychosocial support and health information. Participants in the public health sector reported that their follow-up care is down referred to a local day-hospital, meaning that they have a different team of HCWs than those who treated them initially. Conversely, participants in the private sector had easier access to follow-up care and could maintain a sustained relationship with the same oncologist over a long period of time.

Participants reported managing several long-term side effects from chemotherapy and radiation, such as diabetes, high blood pressure, experiences of weakness, stiffness and numbness, lymphedema, changes in taste and smell, brain fog, hearing loss, and changes in skin colour and hair quality. These long-term side effects were described as having a significant impact on their daily lives. For some participants, these long-term side effects were not communicated by their medical team. Participants in the public sector who reported experiencing progression or potential symptoms of recurrence reported difficulty with re-entry into the system, with doctors not checking folders for histories and delays getting appointments. Pain management across the care continuum, predominantly in the public sector due to more challenging diagnostic pathways, is insufficient, from initial visit to end of life care. Furthermore, end of life care information and death literacy emerged as a gap in the patient care journey.

The findings of the paper constellate around three themes: (1) (de)personalised care; (2) Self-advocacy; and (3) intersecting vulnerabilities.

## (De)personalised care

This theme is broken down into three sub-themes: (1) relational; (2) organisational and structural; and (3) suboptimal care.

**Relational.** Many participants reported experiencing care from HCWs that was characterised by empathy, respect for patient autonomy and psychosocially supportive. Participants in the public healthcare sector mostly reported good experiences with HCWs in tertiary facilities as did participants in the private sector. This was articulated through giving diagnoses with sensitivity and compassion, recognising participants' emotional states and building a longstanding relationship with participants. However, some participants reported experiences with HCWs that undermined their dignity, humanity, and individuality. Participants reported occurrences of HCWs talking about them but not directly to them, bringing in students without introduction or stating their purpose, allowing students to do breast examinations without first talking to the participant about what was going to happen, and not giving sufficient health information across various stages of the care continuum. For example, Jennifer, who accessed care through the private healthcare sector, reported during one of the feedback sessions that she was told that she had two-years to live without giving her further health information regarding end-of-life care and experiences. Other participants noted that they were not given their diagnosis or sufficient information about their care journey. Tinashe noted that doctors within the public healthcare sector were overburdened and not always able to provide sufficient health information, and that patient navigation could be a potential solution if it were to provide additional information and support. However, some participants shared how they consciously subverted this depersonalising dynamic. For example, after not being greeted and told to get changed, Gafsa introduced herself and slowed down the interaction to do proper introductions before the doctor could do any clinical examinations. Depersonalised experiences, then, are repositioned as events that participants have agency over and that they can shift into person-centred experiences. Patients are then not passive receivers of care but can shape how care is given. See participant quotes in Table 1 below.

**Organisational and structural.** Participants' lived experiences were shaped by broader organisational and structural factors that impacted their cancer care. Some participants reported that the public sector hospitals had support structures, such as transport and support to obtain social grants, in place to attend to the broader socioeconomic factors that shape how patients can access healthcare and adhere to treatment.

Within the public healthcare sector, some participants reported delays in receiving appointment dates, test results and treatment, due to the volume of patients that need to be seen. This structural level issue results in patients experiencing increased stress and anxiety during these long waiting periods. Compounding this, some participants attributed the advanced stage at their diagnosis to be related to long waiting periods and delays. Participants actively recognise the broader structural factors that impact theirs' (and others') care journey, which shapes how they reportedly interacted with the health system. For example, Leonie reported that she understood that bed space was limited in the hospital where she had her mastectomy and requested an early discharge to free up space for other patients.

**Table 1. Quotes for relational (De)personalising care experiences.**

|  | Quote |
|---|---|
| Supportive or person-centred care | "I went there with my husband, and when I saw that needle, I don't know what happened, but I started crying and I couldn't speak. I was crying so loud. I was so scared. I'm not scared for needles, never ever. But just to think that needle must go in here, never ever. That doctor said, "[Jamiela], no, we stop everything. Tell me what is wrong." I said, "I am so scared." And she said, "No, you know what? I'm going to do something." She told the other sister, go fetch a… I don't know what the tablet is, but I loved that tablet. I got that tablet, and she's talking to me, and she's talking to me. And I'm looking, and I see the blood. And I hear seven times something going off, almost like a stapler. But I feel nothing. And I'm just somewhere… It's so wonderful." *Jamiela, Breast Cancer, Private and Public, Western Cape* |
| Depersonalised care | Even myself, I don't know how to explain it because the doctor said to me you're on Stage 3B [for cervix]. I said to him, "Could you please explain to me thoroughly?" He said, "No, you're on Stage 3B, it ends like that."" *Ouma B, Breast and Cervical, Private and Public, Gauteng* |
| Reshaping interactions | "For my initial consult with a surgical intern, they called me into the room, and he was looking at my folder. And he's speaking and he's speaking to the folder, and he told, "You can go and change." But he doesn't introduce himself or anything, and he's looking into the folder. I stopped halfway to going to change. I thought to myself, I don't think so. Turned around, and I looked at him and I said, "Hello, I'm Gafsa [surname]." That man got such a shock, and he looked up and he introduced himself. And I said, "Okay, now I'm going to go and change." We had the best relationship after that. Not once did I have to ask him why he's doing something. Pre-op, the injection, he explained the process. He explained what was going to happen to me. And that was the best thing ever that I did. I challenged him and I said, "I'm not a folder. I'm a person. Speak to me."" *Gafsa, Breast, Public, Western Cape* |

Partnerships with NGOs - at public sector healthcare facilities in particular – facilitated the provision of personalised care. The support of NGOs within the public healthcare sector for appointments for mammography and surgery, health education and navigation, and psychosocial support improved patient experiences by shortening waiting times and providing a practical and psychosocial support structure. Psychosocial support through NGO partner organisations was also provided in the private healthcare sector and was noted by participants to be an important support structure.

Conversely, the participants who accessed the private healthcare sector reported that they experienced minimal waiting times and delays. Private healthcare participants reported that they were easily able to get appointments for diagnostic testing and treatment. See participant quotes in Table 2 below.

**Suboptimal care and communication.** A few participants reported that they experienced the care that they received as suboptimal. Sheila, a public healthcare patient, stated that the clinic she first presented to had lost her file, she was frequently moved between health facilities and did not receive optimal care. Sheila is HIV+ and presented symptomatic of cervical cancer. Despite current cervical cancer screening guidelines for HIV+ women, Sheila did not report regular Pap smears, nor was she referred timeously when presenting symptomatically. Sheila directly attributes the advanced stage of her diagnosis to delays in the referral pathway. Bongie reported that after feeling a lump, she reported to her local clinic, which immediately referred her to a Secondary facility. At the Secondary facility, she was sent home, told there was nothing wrong and to wait for two months. There were examples of suboptimal communication in participant narratives, including not communicating potential complications and treatment plans. Stella, a private healthcare patient, reported that the surgeon biopsied her lung and caused a lung puncture. When the nurse pointed out that he had done the wrong thing, he reportedly undermined her. A lung puncture is a potential complication for breast biopsies, but Stella was not made aware of this potential complication before the procedure. While all medical procedures can have complications, an important part of care is to explain this to patients before the procedure, a requirement for true informed consent. Jenny, a private healthcare patient, reported that the nurse told her that she would not need chemotherapy or radiation as her surgery was successful. The doctor later confirmed she would need further treatment. This discrepancy suggests suboptimal communication strategies between HCWs, which can impact patients' care. See participant quotes in Table 3 below.

**Table 2. Organisational and structural (De)personalising care experiences.**

| | Quote |
|---|---|
| Patient support structures | "They're very supportive at [hospital name]. They will give you bus tickets. If you come from the country, they will see that you have your food and there's other people around. But I was now fortunate, my children could bring me with a car, take me up and down and so on, so I didn't use the bus. I didn't need it. It could go to someone else. They also asked if I needed a disability grant, but then my husband, I mos now had a husband, so I said, no, it's mos just for the short term and that." *Inki, Cervical, Public, Western Cape* |
| Overburdened public healthcare facilities and waiting times | "Then I got an appointment and unfortunately their system, because cancer is escalating at [name of hospital], there's always a lot of people. So, everything that you must go through, all the tests that you have to go through, it's every three weeks. They can't just book you. I eventually started in 21 with the tests, the biopsy, the mammogram, the... What was the other one? I also had a radiation injection and a CT scan, all that tests. Eventually, it took me about three months for the tests to be done. That is three months without any treatment already." *Desiree, Breast, Public, Western Cape* |
| NGO Partners | "Then she said, okay, seeing that I have a history of cancer, because my mom died, as I said, within five weeks and her father had cancer as well, so I was very lucky because the waiting period at times is three, four months. But the Pink Lady van - mobile van - were there helping out with their backlog and they could see me the same day, so I had my mammogram." *Galiema, Breast, Public, Western Cape* |
| Private healthcare and no delays | "I'm very grateful to have been treated in private healthcare. Within two days I had my results and it was confirmed as breast cancer. I was referred to the amazing [name], which I'm extremely grateful to. Literally, my pathway from diagnosis to a double mastectomy was within two weeks. I was diagnosed with Early Stage 1A lobular, invasive breast cancer. They discovered another lump in my other breast, so I had a bilateral mastectomy with immediate reconstruction." *Nicole, Breast, Private, Gauteng* |

**Table 3. Quotes for inappropriate care.**

| | Quote |
|---|---|
| Suboptimal care | "I rushed to the clinic. Then when I was there in the clinic, so they checked, so they saw there is a lump. It was 3 centimetres. Then they wrote a letter referring to the hospital. They said I must go to [Secondary] Hospital, so I went there to [Secondary] Hospital. When I was there, the doctor who was helping me, he said it's nothing, I must go back home and then I must come back after two months. So, I went back home because it's the doctor who's talking.... When I went back to the hospital, the lump was now more on that 3 centimetre. Same doctor, he checked up on me and then he was so shocked because the last time he said it's nothing and then they say I must go to do biopsy... Then he said to me, "Joh, this thing of yours, it's scary." … they say I must come back after three weeks for my results. I came back after three weeks. Then after three weeks, the doctor said you have breast cancer. I was so shocked. I was so shocked and scared that I didn't know what to do or say. They wrote a letter again that I must go to [Tertiary] Hospital. I went maybe after, it was three months. … the doctor who was attending me, she said, "Why you only come now? Because you saw your lump on March and then at that time, I think it was May." Then he said, why you only come now?" *Bongie, Breast, Public, Gauteng* |
| *Sub-optimal communication* | "Because when I had all the biopsies, they punctured my lung really badly and I couldn't breathe at all and the nurse in the theatre said to the surgeon, you've done something wrong. He said, oh. He was really annoyed with her... He basically poo-pooed her and did three more biopsies in the same lung. Anyway, the next day when they did all the bone scans and chest X-rays and everything, it's actually my brother-in-law, his brother came through and he said, "Stella, have you been really battling to breathe, get off chairs?" I said, "I haven't been able to get off a chair or out of the bath, Mark had to pick me up." He said, "Look at your lung. Look at what he did to you," and he showed me. They watched me for two weeks. I had to have X-rays every day to make sure the lung doesn't collapse completely." *Stella, Breast, Private, KwaZulu Natal* |

## Self-advocacy

Participants drew on several strategies to advocate for themselves throughout their care continuum. These points of self-advocacy and refusal highlight tension points along the care continuum where patients may receive delayed or suboptimal care, demonstrating that patients are active participants in the shaping of their experiences within the healthcare system. During initial care seeking, public and private sector participants reported advocating for themselves, when providers did not assess their symptoms as serious. Participants reported using different strategies to manage perceived HCW misappraisal of symptoms. Some directly asked for a referral, others immediately sought a second opinion. One

participant reported going directly to the tertiary hospital, bypassing delays in the referral pathway, despite knowing that she needed a referral letter. Some participants without medical aid would bypass the public healthcare sector referral pathway by paying for an initial private GP consultation. Some participants reported advocating for themselves at multiple points in their healthcare journey. Shiela and Sheleen, participants with cervical cancer, reported that they needed to persistently seek care despite being turned away from clinics, perceived inappropriate symptom management, files lost, and waiting for appointments. The strategies outlined by the participants reportedly facilitated a faster move through the referral pathway. Some participants reported being active participants in the selection of the treatment regime, refusing some or all the proposed treatment plan and some seeking alternative opinions. See participant quotes in Table 4 below.

## Intersecting vulnerabilities

Participants' pathways along the cancer care continuum are shaped by intersecting vulnerabilities. One participant resists the idea that she is a "survivor" because of her cancer diagnosis and repositions herself as a survivor of multiple intersecting vulnerabilities, such as socioeconomic status, family dynamics, lack of access to education, and gender.

"…I'm a survivor, because I've never gone to school. I've never attended a school, but I thank God that I can read and write and that He carried me through this. I have a lot of illnesses. I lost a child in my marriage. I am alone, my mom has seven children, all of them are alive, I lost a child in my marriage. I got divorced after that. I have a child in prison. I have a child who lives on the streets, who smokes buttons, a girl, my only girl." *Sheila, Cervical, Private and Public, Western Cape*

**Table 4. Quotes for self-advocacy and refusal.**

| | Quote |
|---|---|
| Requesting referral | "Anyway, I went to the day hospital in [area], not the day hospital, community clinic. The sister couldn't feel the lump when she did the exam, but she believed me when I said that I'd felt something, and she gave me the referral letter to process. So, I went to the breast clinic at [tertiary hospital]. I got to see [name of doctor], amazing young doctor, and she confirmed, yes, it is a lump." *Gafsa, Breast, Public, Western Cape* |
| Seeking second opinions | "Next day, I went to the doctor. Doctor's looking at me. No, there's nothing. You don't have cancer. Straight out. You don't have cancer. Felt, gave me strong ointment, gave me Panados, and gave me antibiotics. Went home. But I don't know why, but you know your body. Somewhere, I don't know where, but there is something in here that tells you something isn't right. Because I never felt the same. I knew something was wrong, and I told my husband, something isn't right. I want to go to, I told him, a real doctor … Then we went to Belhar, went to a doctor there. She told me I must go for a mammogram." *Jamiela, Breast, Private and Public, Western Cape* |
| Bypassing referral processes | "Then I phoned the breast clinic after two weeks. I couldn't get hold of them. The phone would just ring. But I was also told you can't just walk into the breast clinic back then. You had to have a doctor's referral. I thought, no, man. I told my husband to drop me at the breast clinic and I just walked in, and I lied through my teeth. I said I took three taxis to get here and I can't go back home. Can't they just see me? They said, no, I need to go to my GP, and I need to get referred and whatever. I begged this woman. I said, I've struggled for weeks to get through on the phone. If I had gotten through and they told me. So, I'm giving all these excuses. Eventually the oncologist, okay, the go-ahead and she agreed to see me." *Galiema, Breast, Public, Western Cape* |
| Persistence and insistence | "My journey began in 2021. But before that I bled for about four to five years and every time, I was in the day hospital, and out of the day hospital, the doctor just said it was menopause. After a while I said to myself, menopause, but you can't bleed constantly like that for menopause. Then I went to the clinic, then I put myself in [unclear] for three months. Okay, by the last month I went there, I saw the sister again. Then I told the sister, but I'm still bleeding. It's a three-month injection and you bleed throughout the three-month injection, now what does that tell me? Something is wrong. She calls the clinic doctor to come, and he tells me, okay, we're going to do a Pap test. Finally, when the Pap test came, it showed nothing. It was clear. Now, I say, where does the blood come from then? Then he told me, okay, I'm going to make you an appointment for the gynae. That was November 25 for the gynae, 2021. I was in pain, I cried, I didn't know what to do. I lost weight. I couldn't eat. I only cried." *Sheleen, Cervical, Public, Western Cape* |

This section is split into: (1) socioeconomic; (2) gender; (3) comorbidities; and (4) mental health and substance use.

**Socioeconomic.** Socioeconomic lived realities were embedded within participants' recollections of their cancer care journeys. Some participants who accessed the public healthcare sector already had limited access to financial means prior to their cancer diagnosis, reporting unemployment, food insecurity, and no money for transport to healthcare facilities. A cancer diagnosis exacerbated these existing financial challenges by placing additional burden on participants to travel to healthcare facilities. Some participants who had challenges within the public healthcare sector referral system reported seeking out care at private GPs, which had additional financials that some participants could not afford. For some public sector patients, material support from family members facilitated easier access to healthcare, including transport and paying for private GP appointments.

Stories about socioeconomic means were also tied to other intersecting identities, such as aging and retirement, and gender and marital dynamics. Some participants previously had access to medical aid while working or married, but lost access to medical aid due to retirement or divorce.

Some participants reported that their cancer diagnosis resulted in them needing to stop working, and they became reliant on grants for survival for themselves and their families. Social grants that could assist with this financial burden were reported to be insufficient. Ouma B reported an additional barrier to receiving a disability grant. Despite providing the correct document to the South African Social Security Agency (SASSA), she was denied based on not looking "sick" enough. This bureaucratic denial has a material impact on Ouma B's lived reality and acts to further marginalise her. Ouma B spoke about how generational poverty, where herself, her mother and her niece are unemployed, has a profound impact on her cancer care journey.

Conversely, participants who have the socioeconomic means to access medical aid and private healthcare note that they have easy access to health services, such as mammography, psychologists, dieticians, physios and second opinions. See participant quotes in Table 5 below.

**Gender.** At various points in their stories about their cancer care journeys, the participants reflected (in)directly on how being a woman shaped their experiences. Some participants shared that they were still expected to uphold traditional gender roles within their families or to appear "strong" and self-reliant, even when they needed social support.

**Table 5. Quotes for socioeconomic realities.**

| | Quote |
|---|---|
| Financial realities | "And by looking at your surroundings, I used to cry. It happens. At times when you think, Lord, what is happening? I've been fighting cancer, stress at home, this and this, financial problems, food, not having food at home, transport, money transport and what not. All those things, when you pile them together, you just see tears falling from your eyes. Why me? What did I do to deserve such life?" *Nthabi, Breast, Public, Gauteng* |
| Cost of seeking care from private General Practitioners | "Then that doctor told me he was now going to do a Pap smear for me because of the bleeding. Then I said, but I don't have R250 (~14 USD) for the Pap smear, because the Pap smear is R250 and my [consultation]." *Shireen, Cervical, Public, Western Cape* |
| Material support from families | "Then he said, Mommy, just stay there. I'm going to Uber you home now. Don't come home with a taxi like you normally go. They didn't want me to come with the taxi or so." *Shireen, Cervical, Public, Western Cape* |
| Grants | "And you see now, what the doctor did to us, he gave us the SASSA form and then we must take that form to SASSA and give back to him. I feel so bad. When you take those forms, everything, you take it back to SASSA, those doctors from SASSA, they just look at you. Really, are you sick? Really, are you sick? Do you need money for SASSA? They decline it… [SD: How do you cope with not being employed and having trouble with SASSA?] It's very hard. It's very hard because I'm not working and my mother, she's a pensioner. Also, my niece are not working at all. No one is working at home. We live with grant. It's very bad…I gave them everything, my dear. Everything. And then they said to me I must do the appeal after three months. Am I going to survive? How am I going to survive?" *Ouma B, Breast and Cervical, Private and Public, Gauteng* |
| Private healthcare | "By the way, I also went four counselling. Because I felt I wasn't handling the situation, so I saw a psychologist for six sessions…" *Presheen, Breast, Private, KwaZulu Natal* |

Experiences of motherhood shaped some participants' journeys in a myriad of ways. Strikingly across most of the FGDs, conversations about motherhood and childcare brought about strong emotions. Some participants reported that their children were motivators to undergo treatment modalities that they otherwise would not readily choose. Participants recounted disruptions to their ability to perform duties associated with motherhood, such as breast feeding, childcare and physical affection, and separation from their children due to treatment were distressing.

There were mixed experiences with male partners across participants cancer care journeys. While some participants reported that they had a supportive relationship, many others stated that they experienced conflict, separation or divorce because of their cancer diagnosis. For participants with cervical cancer, bleeding and pain during sexual intercourse brought about conflict with their partners. See participant quotes in Table 6 below.

**Comorbidities.** Some participants reported that their cancer care journeys were complicated by comorbidities, such as HIV and TB and autoimmune conditions. Particularly, some participants with cervical cancer who were HIV positive and symptomatic noted that HCWs did not refer timeously. See participant quotes in Table 7 below.

**Mental health and substance use.** Some participants' cancer care journeys are also embedded in contexts on mental health challenges and substance use. See participant quotes in Table 8 below.

**Loss.** Many participants framed their cancer journeys in relation to experiences of loss by cancer, other illness or violence of close family members. Cancer-related loss of family members prior to their own diagnosis was reported to shape participants' perceptions of their own symptoms, how they approached their family about their cancer diagnosis, and how they responded to their cancer diagnosis. A few participants mentioned that they had a family member that passed away due to cancer, but they were unaware that their family member had been diagnosed. Some participants reported that close family members passed away during their treatment, impacting their physical, financial and emotional support structures. See participant quotes in Table 9 below.

**Table 6. Quotes on gender.**

| | Quote |
|---|---|
| The 'strong' woman | "It's a lot because by then I'm going through the divorce and now it's cancer. It's a long journey and you have challenges with your life, with my ex-husband and kids are then and there. My kids are very young, and I can't share it with them, what I'm going through. I have to be strong for them." *Winnie, Breast, Public, Gauteng* |
| Disruptions to 'motherhood' | "I went for the chemo and things started to get better. And so, when I was finished with the chemo, the pain came back, like now. It came back and I couldn't eat, I couldn't walk. I couldn't even talk to anyone. I was just sitting there. I couldn't sleep or lay on my back or so. I was sitting up straight and sleep. I had to send my children away because I couldn't even sit with the baby and [crying] it was a really stressful time for me. I couldn't handle it because my baby was still small. I finished with the treatment, and everything went well and so on." *Chantel, Breast, Public, Western Cape* |
| Experiences with partners | "I went for chemo. I went for the op, but in that time my husband, we split up because while I was doing the tests and that, things wasn't so well between the two of us and so we split up and he left me. When I went for my op, it was like when I was finished, they explained to me it was taken out and the things that I must now to do to look after myself... And the fact that my husband left, and I was alone. My kids were there, my family was there, but at night when I go sleep, I'm alone." *Fatima, Breast, Private, Western Cape* |

**Table 7. Quotes on comorbidities.**

| | Quote |
|---|---|
| Comorbidities | "Anyway, went to have the surgery, but because of the rheumatoid arthritis and the pacemaker, it means that normal surgery for me is not the same as everybody else's because they've got to use different instruments because they can't use magnet things and electro funny things. A surgery that should have maybe taken maybe four hours or whatever took eight hours. It was very, very long and very, very painful." *Jenny, Breast, Private, KwaZulu Natal* |

**Table 8. Quotes on mental health and substance use.**

| | Quote |
|---|---|
| Mental health | "I had a rough upbringing, it's kind of rough because I was emotionally and physically abused by my sister, so I wanted to die. I was taking Rattex. I was popping them like pills. I took thinners, drinking thinners. I've tried slitting wrists and nothing happened. Then there was even a day I was like, it was a low day and it was before my mental space got in check. I even said to my brother, he's my cousin, but he's very close to me so I call him my brother, I was like, even the cancer can't kill me. How am I going to die?" *Tinashe, Breast, Public, Gauteng* |
| Substance use | "I've lost my kids in 2022. I've lost twins. They had to remove my tube. I'm left with one tube. Almost died that day also, but God was there for me. In 2022 December I felt a lump, but because I was drinking heavily because I couldn't... How can I say? I couldn't deal with me being separated from my husband, he got a girl pregnant in our marriage and I couldn't deal with that and I couldn't deal with a lot of things, so I was falling into drugs, started using tik. In December I felt a lump, but I didn't worry because I was heavily drinking and on drugs and all that. In January, I went to the doctor first and then the doctor told me now to come back in a few weeks' time." *Leonie, Breast, Public, Western Cape* |

**Table 9. Quotes on loss.**

| | Quote |
|---|---|
| Cancer-related loss prior to diagnosis | "My grandmom had breast cancer. Then I didn't have a clue, but I knew she was sick and in lots of pain. I used to accompany her to the hospital as there was no one to do that… Seeing her going through all the pain of injections, I remember one day looking at the student doctor making or doing biopsy on her. I couldn't hold back the tears as he repeatedly injected her several. I couldn't take it… She only complained to me when we left. My heart was bleeding… Little did I know that I'm being strengthened for my journey. She then passed away due to breast cancer. I remember praying that God spare her the pain. I couldn't take it no more. No one deserves to be in the thinnest line between life and death… I went and was told that I'm on Stage 2 cancer. [Doctor] advised on cutting both breasts as it was genetic. I agreed and fought for my survival. As a tribute to my late granny, who didn't see the need to fight, maybe she felt she was too old, she had lived so much pain, and I also cursed it. I pray and believe no one in the next generation should go through so much pain." *Nthabi, Breast, Public, Gauteng* |
| Loss during treatment | "I was diagnosed July. 11 August my mother passed on. My mother died the Friday and then I was just blaming myself because I was just like, she died because she couldn't take it that I was diagnosed, and she was worried about what's still going to follow. We went to bury my mother…When I came back, I went for my first chemo. That was so difficult because losing your mother is like the whole world is over. Because I was just like, I still have my parents, that was why I was okay with being diagnosed with breast cancer. I still have my parents, so they will still be there for me and my family and I'm happy… 31 August my father died. I went to go bury my father also. I had to come back, my second chemo… With my father, when we go the hospital, he knows where to go. Then when I went to the hospital, I got lost. I just immediately went on my knees and I started crying that morning…" *Leonie, Breast, Public, Western Cape* |

## Discussion

Patients with breast and cervical cancer in SA have diverse experiences which are embedded within an unequal healthcare system rooted in histories of colonialism and apartheid [43]. Access to and quality of care is still shaped by socio-economic status [44,45] and historical geographic distribution of healthcare infrastructure and resources [46], and have been reinforced by rapid urbanisation, a global shift towards neoliberal policies, inadequate economic growth, and poor management by governance structures [47]. Patient journey mapping as public health method offers an opportunity to redesign a patient responsive healthcare system by pinpointing actionable opportunities along the cancer care continuum to redesign a more patient responsive system. By using patient journey mapping, we were able to plot the complex ways in which patients with breast and cervical cancer move through the public and private healthcare systems and engage with HCWs and other important stakeholders in the system. By examining everyday behaviours of patients and the reported actions of HCWs, we can explore how broader features of the South African healthcare system impact cancer care journeys.

Participants in this study report how accessing cancer care through the public and private healthcare sectors shapes their care journey. However, the public and private sectors are not disparate experiences for patients, as there are

slippages and interconnectedness between these two sectors. While mediated by socio-economic means, participants make active decisions to move between these sectors for different goals. The transition between these sectors varied: some participants initially sought care from private GPs and subsequently moved to public healthcare for diagnostic evaluations and/or treatment. Conversely, others began their care within the public primary healthcare system and later consulted GPs to expedite referral processes back to public services. Some participants exclusively accessed healthcare either through the public or private healthcare sector. Moving between the public and private sectors is a strategy that some participants employed to circumvent challenges within the public sector, such as long waiting times, perceived suboptimal HCW appraisal of symptoms and poor referral processes. Participants in both the public and private sector had trouble in navigating life after diagnosis and treatment. Specifically, once treatment is completed, participants reported a lack of psychosocial support in the public sector while for those on medical aid schemes they still need to pay out of pocket for rehabilitation and psychosocial health services (e.g., occupational therapy, physical therapy, psychologists and so forth) due to depleted funds. NGOs fill an important gap related to health information and psychosocial support throughout the care continuum. Participants recommended additional health information post-treatment on what to expect, long-term side effects, recurrence and required check-ups. Participants with breast cancer in the public sector reported that the compression wear for lymphoedema does not match their skin tones and recommended diverse options.

While scholarship on public perception of the public and private healthcare sectors in SA reports that it tends to be dichotomous, with public positioned as 'bad' and private as 'good' [48,49], participants in this study resist the narrative that the public sector is all 'bad' and provided a nuanced discussion of quality of care provision across the sectors. Participants generally shared positive experiences with public tertiary care, highlighting the quality of treatment and support received, but expressed significant concerns about the primary healthcare sector, particularly around delays, inadequate assessments, and poor communication. Participants emphasised moments of personalised care and ways that HCWs provide excellent care despite structural challenges, including support for social grant applications, good communication and establishing good relationships. Public sector partnerships with NGOs for services, such as mammography and psychosocial support, strengthened provision of quality, holistic care. These moments of humanising care exist within the challenges of long queues at health facilities, poor treatment by some HCWs, long waiting times for appointments and test results, and referral delays [16,50–53]. Patient journey mapping as method offers an opportunity to examine how these tensions impact everyday experiences of patients with breast and cervical cancer. In a study by Harries et al. [52], participants with cervical symptoms reported experiencing judgment from HCWs and the association of symptoms with promiscuity. Conversely, participants with breast symptoms had a better experience. Our study also notes that the journey to diagnosis for participants with cervical cancer took much longer than for those with breast cancer, and they reported experiencing more referral delays. Participants who exclusively accessed the private healthcare sector for their cancer care reported fewer delays in referral and testing. Participants troubled the perception that the private health sector is all 'good' by highlighting moments of poor care provision, suboptimal communication and high care cost [48].

Across both the public and private sectors, participants reported suboptimal communication between HCWs and patients. Optimal communication between HCWs and patients can promote patients' full participation in their care and take responsibility for their health journeys [54]. However, suboptimal communication can undermine the quality of care, patient decision-making and clinical practices [54,55]. Suboptimal communication ranges from a lack of adequate information to verbally abusive communication practices [54]. Other studies conducted in SA have indicated that patients with cancer reported a lack of information provided by HCWs about their referral, diagnosis, treatment and side effects and required follow-up appointments [16,51,52]. Some participants in our study reported that they did not receive sufficient information during their diagnostic workup regarding potential complications, their treatment journeys, and post-treatment care. In the feedback sessions, the participants recommended that, "Doctors must manage us as patients, we are not

numbers". Elaborating further, they stated that healthcare workers need to manage patients holistically and use communication strategies that are empathetic, respectful, transparent and centre consent. Within teaching facilities, participants recommended that healthcare workers talk directly to them rather than amongst themselves. Health information to prepare newly diagnosed patients on how to manage living with cancer should start at the first consultation, and not after treatment consultations. Additionally, they recommend that healthcare workers need to "hear patient voices" through translation of research findings. One example that was given was turning the findings from this study into a graphic novel which can be used by healthcare workers to prepare newly diagnosed patients on how to manage living with cancer.

Our study highlights the urgent need for strategies to improve HCW communication about referral pathways, diagnosis, treatment options, side effects, and potential complications. In their scoping review, Graetz et al. [56] identified several enablers of effective communication in paediatric oncology, which include multidisciplinary family meetings, repetition of information, written information, ongoing caregiver support, the development of a therapeutic relationship with the clinical team, and various tools such as pamphlets, drawings, non-verbal communication and preconsultation lists. Patient navigators and community healthcare workers have also been shown to improve health communication and patient satisfaction throughout the cancer care continuum [57,58]. According to Chan et al. [58], there is strong evidence that patient navigators improve screening rates, shorten pathways to diagnosis, reduces hospital readmissions, improve adherence to surveillance appointments, improve patient satisfaction and quality of life, and improves communication, patient decision making and treatment knowledge. This study demonstrates that patient-centred communication is a central concern when redesigning a patient responsive cancer care continuum.

Patients' journeys through the cancer care continuum are not just impacted by the healthcare system itself but also by intersecting vulnerabilities, such as socio-economic status, gender, comorbidities and mental health and substance abuse. Participants in our study also framed their experience in relation to loss of close family members prior to diagnosis and during and after treatment [59]. Socio-economic challenges were predominantly reported by participants accessing public healthcare. Socio-economic challenges for reaching cancer care in SA are well documented, noting that the financial cost of getting to secondary and tertiary healthcare facilities for diagnosis and treatment is high for patients [51]. Particularly for rural areas, lack of transport or money for transport results in patients having difficulty in accessing health facilities [52]. Additional financial challenges that emerged in our study were loss of income and ability to work and relying on social grants. Our study highlighted some of the strategies that the public healthcare sector has built to protect patients from financial crisis, such as assisting with grant applications. For some participants, families were an important support structure for material support. Within the SA context, many patients with cancer have other comorbid conditions, such as HIV, hypertension and TB, which results in increased contact with PHC facilities, impacts family-patient dynamics and shapes experiences of care [59–61]. Heteronormative gender expectations, such as bearing most of the burden for family care and partner pressure for sexual intimacy, and disruptions to their ability to adhere to these expectations shaped participants' experiences in multiple ways. Partner conflict, separation and divorce were common experiences.

Participants were not passive receivers of cancer care but were active agents who worked to shape healthcare interactions. The study participants recommended that community outreach and education campaigns are needed to improve the public's knowledge of cancer risk factors and symptoms to provide tools for patients to advocate for better care in health facilities and address stigma and misinformation. A study by Johnsons et al. [62] on women's experiences of breast and cervical cancer from Andean countries stated that patients used self-advocacy, persistence and protests at various stages of their care journey, employing strategies such as following up after misdiagnosis, asking questions at appointments, rejecting misinformation and stigma, verbal confrontation when faced with system delays, and making decisions independent of HCWs and family [62]. Coetzee et al. [63] also noted that patients with breast cancer sought appropriate confirmation of their diagnosis and subsequent treatment even when it was not routinely available. The everyday ways in which patients with breast and cervical cancer act to cope, survive or undermine broader systems of power embedded within the healthcare system are important for the reconfiguration of psychosocial, physical and affective spaces in

which marginalised people may exist [64,65]. This study presents examples of ways in which participants navigate and resist challenges in the healthcare system, such as bypassing perceived ineffective referral pathways by attending tertiary facilities without a referral, attending a private GP for a faster referral to public tertiary facilities, slowing the doctor down when proper introductions have not been made, and telling HCWs how they would like to be spoken to. This study also highlights ways in which participants reported that HCWs worked around the system to provide quality of care despite challenges embedded in the system.

This study demonstrates several key strengths, notably its strong patient-centred focus, which addresses a significant gap in South African and broader LMIC cancer care literature by amplifying the voices of breast and cervical cancer patients. The use of interpretive phenomenological analysis and participatory journey mapping methods enabled a rich, nuanced understanding of patients' lived experiences across the cancer care continuum. The study design was ethically sound and iterative, incorporating participant feedback sessions to validate and refine findings after initial data collection. The inclusion of participants from both public and private healthcare sectors across three provinces enhances the diversity and contextual relevance of the data. However, it is important to note that perspectives from patients with breast and cervical cancer from other provinces are missing from this study, and the participants were primarily from urban communities and were able to seek cancer care within the province in which they resided. Rural experiences and pathways are missing from this study. Particularly in KwaZulu Natal, we were not able to recruit patients from the public healthcare sector as our partners did not have existing support groups or advisory groups for patients who have completed treatment. We relied on volunteer participants, who were all private healthcare sector patients, through another NGO working within the area. As the participants had completed treatment at the time of data collection, their reflections may be influenced by retrospective recall biases. Participants receiving end of life group was not well represented because of participants cancelling due to ill health on the day of data collection. As this study intentionally focused on patient experiences, provider perspectives are not included.

## Conclusion

Grounded in intersectionality and health equity frameworks, this study used patient journey mapping – combining individual and collective mapping – to explore how women with breast and cervical cancer navigate and negotiate care in complex and unequal healthcare systems in three provinces in South Africa. Going beyond tracing clinical pathways, this study examined social, economic and affective contours of cancer care and how these factors produce differentiated access, healthcare experiences and lived realities. We traced how patients navigate and negotiate care across public and private sectors, exposing both systemic barriers and practices of humanising care within structural constraints. Within these systems, patients are not passive recipients of care: their narratives demonstrate agency, self-advocacy, and acts of resistance that have the potential to reconfigure interactions with healthcare workers and services. Patient journey mapping is a valuable approach for identifying inequities and actionable opportunities for system redesign. By translating patient experiences into visual and collective insights, it can inform more responsive, equitable and compassionate models of cancer care. Strengthening communication, continuity and support across the care continuum remains central to building a health system that truly reflects and responds to patient realities.

## Acknowledgments

Thank you to the Cancer Association of South Africa (CANSA), Campaigning for Cancer and Reach for Recovery for their assistance with recruiting participants and providing space to conduct data collection.

This study was conducted as part of Working Group 4 for the Lancet Commission on The Lancet Commission on Cancer and Health Systems. We would like to acknowledge Felicia Knaul, Valentina Vargas, Xiaoxiao Jiang, Patricia B. Pedreira and Mike Touchton for their contributions to the protocol development for the study.

## Author contributions

**Conceptualization:** Sarah Day, Alec Payne, Jennifer Moodley.

**Formal analysis:** Sarah Day, Bukeka Sawula, Alec Payne, Shameem Bray, Denis Okova, Jennifer Moodley.

**Funding acquisition:** Sarah Day, Jennifer Moodley.

**Methodology:** Sarah Day, Alec Payne, Jennifer Moodley.

**Project administration:** Sarah Day.

**Supervision:** Sarah Day, Jennifer Moodley.

**Visualization:** Sarah Day.

**Writing – original draft:** Sarah Day, Bukeka Sawula.

**Writing – review & editing:** Jane Harries, Bukeka Sawula, Alec Payne, Shameem Bray, Denis Okova, Lauren Pretorius, Jennifer Moodley.

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
