## [Decision Letter · Decision Letter 0]

7 Sep 2025

Dear Dr. Day,

We look forward to receiving your revised manuscript.

Kind regards,

Sogo France Matlala, PhD

Academic Editor

PLOS ONE

Journal Requirements:

[Cancer Association of South Africa (CANSA) Type B Grant].

[This research was supported by funding from the Cancer Association of South Africa (CANSA).]

[Cancer Association of South Africa (CANSA) Type B Grant]

[SD is an academic editor for PLOSOne. All other authors have no competing interests to declare.].

5. In the online submission form, you indicated that [Data cannot be shared publicly because of issues of confidentiality. Data can be made available at reasonable request.].

6. Please amend the manuscript submission data (via Edit Submission) to include author Sarah Day.

7. Please amend your authorship list in your manuscript file to include author Sarah Kathrine Day.

8. Your ethics statement should only appear in the Methods section of your manuscript. If your ethics statement is written in any section besides the Methods, please move it to the Methods section and delete it from any other section. Please ensure that your ethics statement is included in your manuscript, as the ethics statement entered into the online submission form will not be published alongside your manuscript.

9. We note that Figure 1 in your submission contains copyrighted images. All PLOS content is published under the Creative Commons Attribution License (CC BY 4.0), which means that the manuscript, images, and Supporting Information files will be freely available online, and any third party is permitted to access, download, copy, distribute, and use these materials in any way, even commercially, with proper attribution. For more information, see our copyright guidelines: http://journals.plos.org/plosone/s/licenses-and-copyright.

10. We note that your paper includes detailed descriptions of individual patients/participants. As per the PLOS ONE policy (http://journals.plos.org/plosone/s/submission-guidelines#loc-human-subjects-research) on papers that include identifying, or potentially identifying, information, the individual(s) or parent(s)/guardian(s) must be informed of the terms of the PLOS open-access (CC-BY) license and provide specific permission for publication of these details under the terms of this license. Please download the Consent Form for Publication in a PLOS Journal (http://journals.plos.org/plosone/s/file?id=8ce6/plos-consent-form-english.pdf). The signed consent form should not be submitted with the manuscript, but should be securely filed in the individual's case notes. Please amend the methods section and ethics statement of the manuscript to explicitly state that the patient/participant has provided consent for publication: “The individual in this manuscript has given written informed consent (as outlined in PLOS consent form) to publish these case details.

Reviewers' comments:

Reviewer's Responses to Questions

**Comments to the Author**

1. Is the manuscript technically sound, and do the data support the conclusions?

Reviewer #1: Yes

Reviewer #2: Yes

2. Has the statistical analysis been performed appropriately and rigorously?

Reviewer #1: N/A

Reviewer #2: N/A

3. Have the authors made all data underlying the findings in their manuscript fully available?

Reviewer #1: Yes

Reviewer #2: No

4. Is the manuscript presented in an intelligible fashion and written in standard English?

Reviewer #1: Yes

Reviewer #2: Yes

Reviewer #1: Navigating Cancer: Insights from Patient Journey Mapping

Thank you for the opportunity to review this interesting and important topic. Research shows that cancer is the world’s biggest killer, with 10 million deaths per year due to the disease.

Comments

Abstract

The abstract does provide a background and introduction of the study.

Main study

Introduction

The presence of important and current background information on the study topic is noted and appreciated.

Methods

Mapping patient journey is an interesting and unique research design appropriate to understand patients’ cancer journeys across the continuum, highlight disparities, and gaps in access and care. Sampling, data collection instruments, data collection and analysis methods have been clearly described and applied well to yield the reported results. However, researchers are not explicit as to how participant recruitment and data collection was done in the three different provinces. How it is written can confuse the reader, e. g, “map of the municipality where data collection took place” can make one think data collection took place in one province and one municipality.

Ethical considerations

Ethical considerations pertaining to person covered principles of autonomy, informed consent, anonymity and confidentiality. However, pertaining to institution, only ethical clearance from a university and nothing is mentioned about approval by provincial gatekeepers.

Results

Clearly and logically articulated to facilitate repeatability.

Discussion & implication

Discussion relates results to available literature and is placed well within the context of existing research including what is happening in the SA health system.

Conclusions

This section reflects on the discussion section above.

References

Sources have been appropriately referenced, in text and in the list of references at the end.

Recommendation

Based on the above comments, it is recommended that this paper be accepted with minor corrections.

Reviewer #2: Thank you for the opportunity to review this beautifully written paper. The authors have provided a very clear description of the cancer situation in South Africa, the patient journey mapping process they chose, underpinned by a strengths based narrative.

The very clear description of the process of individual mapping, collective mapping, producing a collective visual journey map, coding and theme development using interpretive phenomenological analysis nVivo, informed by theoretical underpinnings of intersectionality and health equity makes this paper a very important paper in the emerging and growing field of journey mapping, and also in cancer care.

Introduction provides a clear picture of the situation relating to breast and cervical cancer in South Africa, and the use of journey mapping to explore the cancer journeys of people internationally.

Research design is well described and the SA health system context provided. Is very helpful for an international readership. The theoretical underpinnings of intersectionality and health equity give the study and paper depth, and as discussed moves from a uni-dimensional view of their lived experiences to a multidimensional nuanced understanding of their cancer journey.

The description of the pilot study and participant preferences is very helpful for other studies. The three step process undertaken in focus groups is clearly and clearly explained (it sounds very effective).

The analysis process and identification of codes and themes using interpretive phenomenological analysis and NVivo is well described.

Findings – very comprehensive.

Line 258 – re pain management being insufficient – was this for both public and private patients?

Figure 1 – very effective visual summary

Discussion

The authors could perhaps include more emphasis on specific recommendations fro the women, or themselves for health system improvement in the discussion. (optional due to word length)

Conclusion – I encourage the authors to lengthen the conclusion by a few sentences to more fully emphasise the key points they would like the reader to take away. This might include the voices, the patient journey mapping method incorporating both individual and collective mapping and emerging themes, underpinned by theoretical underpinnings of intersectionality and health equity to identify both strengths and limitations of health care interactions, services and systems.

**Do you want your identity to be public for this peer review?** For information about this choice, including consent withdrawal, please see our Privacy Policy

Reviewer #1: No

Reviewer #2: No

---

## [Author Response · Author response to Decision Letter 1]

5 Dec 2025

We have provided a reviewer response table as an additional attachment, responding to each of the Editors and Reviewer’s comments.

---

## [Decision Letter · Decision Letter 1]

9 Mar 2026

Navigating Cancer: Insights from Patient Journey Mapping

PONE-D-25-30735R1

Dear Dr. Day,

We’re pleased to inform you that your manuscript has been judged scientifically suitable for publication and will be formally accepted for publication once it meets all outstanding technical requirements.

Kind regards,

Taiwo Opeyemi Aremu, MD, MPH, PhD

Academic Editor

PLOS One

Additional Editor Comments (optional):

Reviewers' comments:

Reviewer's Responses to Questions

**Comments to the Author**

Reviewer #2: All comments have been addressed

2. Is the manuscript technically sound, and do the data support the conclusions?

Reviewer #2: Yes

3. Has the statistical analysis been performed appropriately and rigorously?

Reviewer #2: N/A

4. Have the authors made all data underlying the findings in their manuscript fully available?

Reviewer #2: No

5. Is the manuscript presented in an intelligible fashion and written in standard English?

Reviewer #2: Yes

Reviewer #2: Thank you for the opportunity to review the revised version of this manuscript.

This paper clearly describes the research approach, use of journey mapping, findings, discussion and conclusion, adding to both journey mapping and cancer literature.

Changes to the manuscript have responded to reviewer comments.

Removal of original Table 1 provides de-identification of participants.

Line 297 – check table number (says 9, should be 1).

The added sections in Discussion identifies recommendations from participants more strongly. I considered whether they are findings or discussion but think they fit well in discussion.

The re written conclusion is more powerful.

Acknowledgement inclusions are important.

Conflict of interest statement wording could be altered slightly. Perhaps – ‘we maintained a strong adherence to PLOS ONE policies on sharing data and materials.’

**Do you want your identity to be public for this peer review?** For information about this choice, including consent withdrawal, please see our Privacy Policy

Reviewer #2: **Yes:** Janet Kelly

---

## [Editor Report · Acceptance letter]

PONE-D-25-30735R1

PLOS One

Dear Dr. Day,

I'm pleased to inform you that your manuscript has been deemed suitable for publication in PLOS One. Congratulations! Your manuscript is now being handed over to our production team.

Kind regards,

on behalf of

Dr. Taiwo Opeyemi Aremu

Academic Editor

PLOS One